# Co-Targeting ErbB Receptors and the PI3K/AKT Axis in Androgen-Independent Taxane-Sensitive and Taxane-Resistant Human Prostate Cancer Cells

**DOI:** 10.3390/cancers14194626

**Published:** 2022-09-23

**Authors:** Samusi Adediran, Linbo Wang, Mohammad Afnan Khan, Wei Guang, Xiaoxuan Fan, Hancai Dan, Jianfei Qi, Steven M. Jay, France Carrier, Arif Hussain

**Affiliations:** 1University of Maryland Marlene and Stewart Greenebaum Comprehensive Cancer Center, Baltimore, MD 21201, USA; 2Department of Microbiology and Immunology, University of Maryland School of Medicine, Baltimore, MD 21201, USA; 3Department of Pathology, University of Maryland School of Medicine, Baltimore, MD 21201, USA; 4Department of Molecular Biology and Biochemistry, University of Maryland School of Medicine, Baltimore, MD 21210, USA; 5Fischell Department of Bioengineering, University of Maryland, College Park, MD 20742, USA; 6Department of Radiation Oncology, University of Maryland School of Medicine, Baltimore, MD 21201, USA; 7Baltimore VA Medical Center, Baltimore, MD 21201, USA

**Keywords:** prostate cancer, drug resistance, cell signaling

## Abstract

**Simple Summary:**

Advanced prostate cancer that has progressed after standard therapies such as hormone therapy and taxane-based chemotherapies is an invariably lethal disease state with limited treatment options. There remains an important need to continue to identify new treatment approaches for such patients. We used two cell culture models of prostate cancer that are resistant to hormonal therapy and chemotherapy, and which also manifest some characteristics that are often associated with advanced prostate cancer, such as neuroendocrine differentiation, to evaluate the potential anti-cancer effects of targeting the key molecules, ErbB receptors and AKT. Using several complementary approaches, we found that the concurrent targeting of ErbB receptors and AKT with specific inhibitors was more effective than targeting each of them individually, independent of the underlying molecular characteristics or relative degrees of resistance to the taxanes that defined the prostate cancer models that were studied. Enhanced anti-tumor responses occurred both in vitro and in vivo with dual targeting, with the consistent inhibition particularly of AKT occurring in both settings. These studies provide a framework to evaluate the role of signal pathway modulation as a potential therapeutic strategy in treatment-refractory prostate cancer.

**Abstract:**

Using two representative models of androgen-independent prostate cancer (PCa), PC3 and DU145, and their respective paclitaxel- and docetaxel-resistant derivatives, we explored the anti-tumor activity of targeting the ErbB receptors and AKT using small-molecule kinase inhibitors. These cells manifest varying degrees of neuroendocrine differentiation characteristics and differ in their expression of functional PTEN. Although the specific downstream signaling events post the ErbB receptor and AKT co-targeting varied between the PC3- and DU145-lineage cells, synergistic anti-proliferative and enhanced pro-apoptotic responses occurred across the wild-type and the taxane-resistant cells, independent of their basal AKT activation state, their degree of paclitaxel- or docetaxel-resistance, or whether this resistance was mediated by the ATP Binding Cassette transport proteins. Dual targeting also led to enhanced anti-tumor responses in vivo, although there was pharmacodynamic discordance between the PCa cells in culture versus the tumor xenografts in terms of the relative activation and inhibition states of AKT and ERK under basal conditions and upon AKT and/or ErbB targeting. The consistent inhibition, particularly of AKT, occurred both in vitro and in vivo, independent of the underlying PTEN status. Thus, co-targeting AKT with ErbB, and possibly other partners, may be a useful strategy to explore further for potential therapeutic effect in advanced PCa.

## 1. Introduction

Hormonal therapies and taxane-based cytotoxic chemotherapies represent the major treatment modalities for advanced prostate cancer (PCa). Among the various taxanes in clinical use, paclitaxel was evaluated first in patients with metastatic castration resistant prostate cancer (mCRPC) [1]. Subsequent trials established docetaxel and cabazitaxel as the chemotherapeutic agents of choice in mCRPC treatment [2,3,4,5].

Eventually, both hormonal therapy and chemotherapy fail in patients with advanced PCa. Regarding taxanes, the inherent resistance or the development of resistance during treatment (i.e., acquired resistance) represent major challenges that limit the therapeutic benefit of these agents. Further, the hormone- and chemotherapy-resistant clinical phenotypes that emerge among patients that are treated with standard hormonal and chemo/cytotoxic therapies can display a highly aggressive and invariably lethal clinical course that is collectively being recognized as aggressive variant prostate cancer (AVPC) [6,7,8,9,10]. AVPC includes a spectrum of treatment-refractory clinical disease states that may or may not continue to express varying levels of the androgen receptor (AR) and/or display various gradations of neuroendocrine differentiation (NED). For patients with hormone-insensitive and chemotherapy-resistant PCa, further treatment options are particularly limited, including in terms of immunotherapy-based approaches. There remains a dire need to continue to explore and identify treatment strategies, including in pre-clinical models that reflect aggressive tumor biology, as a step towards developing clinically meaningful approaches for these disease states.

Although no one specific pre-clinical model can recapitulate the complex biology of PCa, two well-studied human prostate cancer cell lines, PC3 and DU145, provide useful context with respect to some of the underlying and potentially relevant molecular characteristics of clinical PCa. PC3 and DU145 cells are AR negative, and although they do not represent the AR-expressing PCa phenotypes, both are androgen-independent and express, to varying degrees, several NED-associated proteins, such as neuron specific enolase (NSE), chromogranin A (CgrA), TUBB3 and others [11,12,13,14,15,16,17,18]. As noted, NED-associated features are an increasingly relevant phenotype during the clinical progression of PCa. Building on the large body of work that has been conducted by others, in the present study, we used PC3 and DU145 cells, and their paclitaxel- and docetaxel-resistant derivatives, to explore whether targeting specific nodes within the defined proliferative signals may be a potential treatment strategy in the androgen-independent/chemotherapy-resistant PCa niche.

Specifically, we focused on the ErbB receptor tyrosine kinase (RTK) family and related downstream signaling, an area of intense investigation that has led to important advances in cancer therapy [19]. For instance, certain mutations in ErbB1 (EGFR) and the amplification/overexpression of ErbB2 provide important targets for interrogation in non-small cell lung cancer (NSCLC) and breast cancer, respectively [20,21,22]. In PCa, although such alterations in EGFR and ErbB2 are generally not observed, ErbB-mediated signaling may play a significant role in the clinical progression and metastasis of PCa [23,24]. Therefore, ErbB homo- and heterodimer-mediated signaling events potentially represent important therapeutic targets [19,25,26].

Major downstream effectors of ErbB are the PI3K-AKT-mTOR-S6 kinase and RAF/MEK/ERK axes that modulate the anabolic and proliferative signals [27]. The PI3K-AKT pathway, in particular, is often activated in advanced PCa via the loss of PTEN, which is a negative regulator of PI3K-mediated AKT activation [28]. Further, studies have established negative cross talk between PI3K-AKT and ErbB3, so that PI3K or AKT inhibition can lead to ErbB3 activation and which in turn may contribute to a resistance to such inhibitors [29,30]. Thus, it has become increasingly apparent that the combined targeting of ErbB receptors and downstream effectors such as PI3K or AKT has the potential to result in greater anti-tumor effects than single agent therapies do [29,30,31]. Given this context, in the present study, we used PC3 cells (PTEN null) and DU145 cells (wild-type PTEN background) and their respective taxane-resistant derivatives to evaluate the role of targeting ErbB receptors and AKT under basal conditions and in response to pathway activation [32]. Using lapatinib and MK2206 as the representative examples of ErbB and AKT inhibitors, respectively, we found that dual targeting leads to: (a) synergistic anti-proliferative, pro-apoptotic responses in both models, and (b) these effects are independent of their underlying PTEN status and basal AKT activation states or relative degrees of resistance to paclitaxel or docetaxel. Our studies provide a framework to explore further approaches that co-target AKT and ErbB in the relatively under-studied area of chemotherapy-resistant NE-type PCa.

## 2. Results

### 2.1. Wild-Type and Taxane-Resistant Prostate Cancer Cells

The PC3 and DU145 cells were selected for their resistance to paclitaxel and docetaxel, respectively, and the resulting cells are designated PC3/Pac20 (PC3/Pac, hereafter) and DU145/Doc60 (DU145/Doc, hereafter), as described previously [33]. Table 1 lists the IC50 values of the wild-type and the taxane-resistant cells for paclitaxel and docetaxel. DU145/Doc cells are highly resistant to paclitaxel (100-fold) and docetaxel (500-fold), and overexpress the ATPase Binding Cassette (ABC) family of transporter proteins PGP and MRP which have been implicated in taxane resistance (Figure 1; [33]). By contrast, PC3/Pac cells have lower levels of resistance to the taxanes (3- to 11-fold) and they do not overexpress either PGP or MRP (Table 1, Figure 1), suggesting that mechanisms other than PGP or MRP likely mediate the taxane resistance in these cells. The fact that PGP and MRP are overexpressed in some but not other PCa models of taxane-resistance is consistent with the observation that ABC transporters account for only a proportion of the taxane resistance that is observed clinically [34,35]. Table 1 also shows the IC50 values of MK2206 and lapatinib for the PCa cells.

### 2.2. ErbB Axis in Wild-Type and Taxane-Resistant Cells

The status of AKT and ERK in the PCa cells in culture under basal cell growth conditions and in response to the EGF treatment (50 ng/mL, 10 min) are shown in Figure 2A,B, respectively. Under the basal conditions, there is minimal pAKT expression in the DU145 and DU145/Doc cells, whereas both cell lines express higher levels of pERK when they are compared to the corresponding PC3-lineage cells, perhaps reflecting some degree of ‘compensation’ amongst the former (i.e., DU145-lineage) cells given their minimal basal activation state of AKT (Figure 2A). Interestingly, although exogenous EGF can lead to the further recruitment of activated ERK (i.e., increased pERK levels) in both PC3- and DU145-lineage cells, and of AKT (i.e., increased pAKT) in DU145-lineage cells, the pAKT levels remain essentially unchanged beyond their already high basal levels in response to EGF in the PC3 and PC3/Pac cells (Figure 2B). This suggests that PTEN loss leads to some degree of uncoupling between the upstream activation of ErbB and the downstream PI3K-AKT axis in PC3 and PC3/Pac cells.

Figure 2C,D shows the response of the PCa cells to the EGF (50 ng/mL) treatment with respect to downstream signaling events over time. The binding of EGF to EGFR leads to the latter’s homodimerization and to EGFR/ErbB2 and EGFR/ErbB3 heterodimerizations, followed by their phosphorylation and activation and that of the downstream molecules AKT and ERK. As reported previously by others, EGF activates the ErbB axis after a short exposure to it (10 min), with the subsequent decay of the activated signals occurring over time. The overall patterns of signal activation and decay between the PC3 and PC3/Pac cells (Figure 2C) and between the DU145 and DU145/Doc cells (Figure 2D) appear to be similar. Taken together, the above data indicate that the ErbB, AKT and ERK pathways are preserved and operative in the taxane-adapted cells as they are in the parental lines; thus, these pathways are potentially useful targets not only in the wild-type but also the drug-resistant PCa cells.

### 2.3. AKT and ErbB Inhibition: Impact on Cell Proliferation and Cell Death

The effects of targeting ErbB receptors with lapatinib and AKT with MK2206 on cell proliferation were evaluated by the Chou-Talalay combination index (CI) method, and the data are summarized in Table 2 [36]. The CI values for the combination treatment are less than one across all the four cell lines; thus, lapatinib with MK2206 leads to synergistic anti-proliferative responses, and these effects are independent of any underlying resistance to the taxanes (Table 2).

The kinetics of single and dual agent targeting on cell proliferation are shown in Figure 3A. The lapatinib/MK2206 combination leads to a greater anti-proliferative response than either agent alone does, with maximal inhibition becoming apparent within 48 h and maintained through day five of the combined drug exposure test. Two dimensional clonogenic assays looking at cell colonies after even longer drug exposures (10 days) corroborate the efficacy of the combination treatments, including in the drug-resistant cells (Appendix A). As expected, treating wild-type PC3 or DU145 cells with taxanes for 24 h results in the accumulation (arrest) of the treated cells in the G2/M phase of the cell cycle (Figure 3B). The PC3/Pac and DU145/Doc cells (which are maintained in taxanes), on the other hand, are not arrested in G2M, but rather, they display a cell cycle distribution pattern that is similar to their respective untreated parental counterparts (Figure 3B); these observations are consistent with the taxane-selected/adapted cells’ resistance to taxane inhibition. In terms of the cell cycle response to the kinase inhibitors, treatment with MK2206, lapatinib, and particularly MK2206 + lapatinib for 24 h leads to the relative arrest of both the wild-type and the drug-resistant cells in G1/G0 (Figure 3C, Appendix A).

The flow cytometry that was conducted using an annexin V/propidium iodide (PI) staining technique shows a trend towards enhanced programmed cell death (PCD; early + late apoptosis) with the drug combinations that were used compared to a single-agent treatments in both of the wild-type and the taxane-resistant cells (Figure 3D). Enhanced PCD after the combination drug treatments is also demonstrated by the live/dead assay that was performed using an acridine orange/PI staining technique (Figure 3E). A trend towards increased PARP cleavage and Bax/Bcl2 ratios in both the wild-type and the taxane-resistant cells with MK2206 plus lapatinib (Figure 3F,G) is consistent with the annexin V/PI flow cytometry and live/dead assay data.

### 2.4. Signal Pathway Targeting in PC3 and PC3/Pac Cells

Downstream signaling events in the PC3 and PC3/Pac cells in response to targeting ErbB and AKT are shown in Figure 4. We assessed the effects of such targeting under basal conditions and after the stimulation of the cells with EGF. MK2206 effectively inhibits the highly phosphorylated and largely EGF stimulation-independent AKT in the PC3 and PC3/Pac cells (Figure 4A,B; lanes 1–4). As expected, MK2206 does not inhibit the EGF-mediated recruitment (i.e., phosphorylation) of ERK (Figure 4A,B; lanes 1–4). As has been well documented by others, a compensatory increase in the ErbB3 expression and ErbB3 phosphorylation occurs in the MK2206-treated PCa cells (Figure 4A,B; lanes 3 and 4), which is consistent with the known inhibitory action of AKT on the ErbB3 transcription factor FOXO. Thus, although MK2206 inhibits AKT, the signaling to ERK via ErbB (ErbB3-containing heterodimers) continues to occur in both of the PC3 and PC3/Pac cells, and which may decrease the therapeutic efficacy of AKT inhibition.

Although ErbB3 is not a direct target of lapatinib, by inhibiting EGFR and ErbB2, lapatinib also has the potential to prevent the ErbB3 trans-phosphorylation that normally occurs when the kinase-deficient ErbB3 forms heterodimers with either EGFR or ErbB2 [37,38,39]. Indeed, lapatinib not only inhibits EGFR and ErbB2, but also ErbB3 phosphorylation, as well as the relative phosphorylation of downstream ERK, in both of the PC3 and PC3/Pac cells (Figure 4A,B; lanes 5–8). However, given the relative uncoupling between ErbB and AKT in these PTEN-null cells, lapatinib does not significantly affect the phosphorylation status of activated AKT in either of the PC3 or PC3/Pac cells (Figure 4A,B; lanes 5 and 6). Thus, lapatinib alone cannot inhibit the activated AKT, while MK2206 alone cannot inhibit the ErbB-ERK axis. Rather, it is the combination of lapatinib and MK2206 that together are more effective in inhibiting both of these pathways (Figure 4A,B; lanes 7 and 8).

### 2.5. Signal Pathway Targeting in DU145 and DU145/Doc Cells

Due to the expression of wild-type PTEN in the DU145 model, there is minimal basal expression of pAKT in these cells. Rather, the ligand-receptor engagement between EGF and ErbB is required for the activation of AKT via the ErbB-PI3K-AKT axis, and of ERK via the ErbB-RAF-MEK-ERK axis (Figure 2; Figure 5A,B, lanes 1 and 2). MK2206 is effective in blocking AKT phosphorylation in the DU145 and DU145/Doc cells (Figure 5A,B, lanes 3 and 4), whereas, the ErbB-ERK pathway remains intact in these MK2206-treated cells (Figure 5A,B, lanes 3 and 4). This provides a rationale for also targeting the ErbB-ERK axis in the MK2206-treated PCa cells (Figure 5A, one should compare the lanes 3 and 4 vs. lanes 7 and 8 in DU145 cells, respectively; Figure 5B, one should compare the lanes 3 and 4 vs. lanes 7 and 8 in DU145/Doc cells, respectively). It should be noted, however, that since ErbB can signal to both AKT and ERK in the DU145 model, lapatinib alone may block activation of AKT and ERK in these cells (Figure 5A,B, one should compare lanes 1 and 2 vs. lanes 5 and 6, respectively). Thus, in principle, a single agent lapatinib, or a more potent pan ErbB inhibitor, should be effective in those PCa cells with a functional PTEN background. Nevertheless, our data overall demonstrate that even in the context of wild-type PTEN function, the dual targeting of AKT and ErbB leads to greater anti-tumor responses in DU145 and DU145/Doc cells (Table 2, Figure 3, Appendix A).

### 2.6. Anti-Tumor Response of ErbB and AKT Targeting In Vivo

The effects of targeting ErbB and AKT were also evaluated in nude mice that were bearing PCa xenografts (Figure 6). We encountered significant difficulty in growing both of the PC3/Pac and DU145/Doc xenografts in the nude mice. Therefore, we conducted in vivo work primarily with the wild-type PC3 and DU145 cells. We evaluated four treatment groups, with seven or eight mice being included per treatment group: Group 1—vehicle-treated (control); Group 2—MK2206-treated; Group 3—lapatinib-treated; Group 4—MK2206 + lapatinib-treated. Compared to the vehicle or single agent treatments, the combination of MK2206 + lapatinib led to a statistically significant decrease in the tumor growth in both of the PC3-based and the DU145-based xenografts, as assessed by tumor volume measurements that were taken over time (Figure 6A,B) and tumor weights at the end of the experiment (Figure 6C,D), which was consistent with the in vitro data. The combination treatments overall were well tolerated, as reflected by the mouse body weights, with a total body weight loss amongst the combination treated mice being within 10% of the range of the controls (Appendix A).

We also evaluated the status of AKT and ERK among five randomly selected tumors from each treatment group (Figure 6E,F). Several observations from these studies are noteworthy. Among the PC3 xenografts (Figure 6E), we noted the following: (a) all control (vehicle-treated) tumors expressed pAKT, as expected, and also to varying degrees, pERK; (b) in the lapatinib-treated tumors there appeared to be minimal inhibitory effects on ERK phosphorylation; (c) in the MK2206-treated tumors, as expected, AKT phosphorylation was inhibited; (d) in the MK2206+lapatinib-treated tumors, it was primarily AKT but not ERK phosphorylation that was consistently inhibited. Among the DU145 xenografts (Figure 6F): (a) not only did all of the control (vehicle-treated) tumors express pERK (similar to DU145 cells in culture), but interestingly, the tumors also expressed pAKT (albeit at lower levels than those in the PC3 tumors), in contrast to the DU145 cells in culture; (b) the lapatinib-treated tumors continued to express pERK; (c) in the MK2206-treated tumors, pAKT was inhibited; (d) in the MK2206+lapatinib-treated tumors, AKT but not ERK phosphorylation was inhibited.

Taken together, the above data suggest that potential discordance in the relative basal activation states of intracellular signaling hubs may occur within the same tumor models under different environments (e.g., in vitro vs. in vivo growth conditions). Under the basal conditions, the DU145 xenografts expressed pAKT, whereas DU145 cells in culture required exogenous EGF to demonstrate activated AKT in the Western blot tests. PC3 xenografts express pERK, whereas PC3 cells in culture show minimal pERK expression without undergoing exposure to exogenous EGF. Although the above treatments lead to the effective inhibition primarily of AKT, they did not do the same for ERK phosphorylation in vivo, nevertheless, as noted, the MK2206 plus lapatinib combination is associated with greater anti-tumor responses both in vitro and in vivo than can be achieved with either agent alone.

## 3. Discussion

The present work builds on the extensive studies that have been conducted by other investigators on co-targeting ErbB receptors and one of the key downstream effector pathways mediated by the PI3K-AKT axis, specifically AKT. Our aim was to evaluate the role of this dual-targeting approach not only in wild-type but also in taxane-resistant PCa models where, to date, there has been a relative paucity of studies on signal pathway modulation as a potential treatment strategy.

The success that has been achieved of targeting the ErbB receptors in some other cancer types has yet to be realized for those of PCa. For instance, in NSCLC, specific mutations in the EGFR can render them susceptible to EGFR targeting [20,21]. In breast cancer, ErbB2 amplification/overexpression allows for the effective targeting of ErbB2 [22]. Such alterations in the ErbB family are not observed in PCa. Interestingly, ErbB4 is generally not expressed in PCa cells, whereas the kinase-deficient ErbB3 serves as an important modulator of ErbB-mediated signaling in PCa, including in mediating the cross talk with the PI3K-AKT axis. Hence, the targeting the ErbB family and relevant downstream events remain potentially viable approaches to explore in PCa, including in settings where the cancer is refractory to the traditional hormonal therapies and chemotherapies.

As small-molecule RTK inhibitors primarily target intracellular kinase domains, direct inhibitors for ErbB3 are not available. Rather, the strategy to target ErbB3 is via the ErbB3-directed monoclonal antibodies or affibodies, or perhaps the small-molecule kinase inhibitors (lapatinib, dacomitinib, among others) that inhibit the other kinase-proficient ErbB members (e.g., ErbB1, 2, 4); such inhibitors can block the activity of ErbB3 by blocking the active hetero-dimer complexes that are necessary for optimal ErbB3-mediated signaling [40,41,42,43]. In this proof-of-principle study, we used lapatinib to target the ErbB family, and MK2206 to target AKT in treatment-resistant androgen-independent PCa cells.

Since the IC50 values for MK2206 and lapatinib are similar between the highly taxane-resistant PGP/MRP-overexpressing DU145/Doc cells and the wild-type DU145 cells (approximately 15 µM for MK2206, 2–3 µM for lapatinib, Table 1), this suggests that these small-molecule kinase inhibitors are not the substrates for the ABC transporters. The MK2206 and lapatinib IC50 values for the PC3 and PC3/Pac cells are also similar between these two cell types (approximately 6 µM for MK2206, 3 µM for lapatinib, respectively; Table 1). However, the PC3-lineage cells are 2.5-fold more sensitive to MK2206 than the DU145-lineage cells are; the greater sensitivity of the PC3 and PC3/Pac cells to AKT inhibition likely reflects their higher basal AKT activation state due to the underlying PTEN loss. Taken together, the data suggest that neither the relative degree of resistance to taxanes (low or high), nor the underlying mechanism(s) of resistance to taxanes (whether PGP/MRP-mediated or not) affect the relative growth inhibition response of the PCa cells to the AKT or ErbB-small-molecule kinase inhibitors, thus making such inhibitors potentially attractive agents to use to target both wild-type and chemotherapy-resistant PCa.

Overall, our studies demonstrate that ErbB-mediated signaling events are largely intact in the taxane-resistant PCa cells, and essentially mimic that of their respective wild-type parental lines (Figure 2). These observations therefore provide a rationale for signal pathway modulation, not only in taxane-sensitive, but also in taxane-resistant PCa cells (Figure 4 and Figure 5). We find that co-targeting ErbB and AKT, rather than each individually, leads to greater anti-proliferative, pro-apoptotic responses in the two independent models of PCa that were examined, including their taxane-resistant derivatives (Figure 3, Table 2). Although PTEN loss (such as in PC3 and PC3/Pac cells) leads to an activated state of AKT under basal conditions, and potentially renders the cells as more susceptible to AKT inhibition (Table 1), AKT and ErbB co-targeting leads to synergistic anti-cellular responses in the PCa cells, independent of their constitutional PTEN status (Table 2). Thus, AKT-based targeting may not necessarily have to be restricted to PTEN minus PCa cells, particularly when they are used in conjunction with effective co-targeting partners.

The enhanced anti-tumor responses with the lapatinib and MK2206 combination among the PCa cells in culture were also observed in the PCa xenografts (Figure 6A–D). Interestingly, we noted a discordance with respect to the relative basal phosphorylation status of ERK and AKT between the cell culture and xenograft models. That is, both pAKT and pERK were expressed in the control DU145 xenografts (Figure 6F), but only pERK was expressed to any significant extent in the DU145 cells in culture (Figure 2 and Figure 5). Similarly, pAKT and pERK were expressed in the PC3 xenografts (Figure 6E), but it was primarily pAKT that was expressed in the control PC3 cells in culture (Figure 2 and Figure 4). In vivo, AKT phosphorylation appears to be most effectively inhibited in both of the PC3 and the DU145 models by MK2206 or the MK2206 + lapatinib doublet, whereas lapatinib when used singly or in combination does not lead to any appreciable and consistent inhibition of ERK phosphorylation (Figure 6E,F). The tumor microenvironment and associated signaling in vivo are likely to be much more complex than that which we can discern using these cell culture experiments, including signaling events in response to defined ligands such as EGF. In vivo, the tumors are likely to be exposed to many paracrine factors and perhaps other growth factors that may be produced by the xenografted tumors. Thus, the activation of AKT and ERK in vivo may also involve events that are beyond ErbB. The feedback recruitment of RTKs other than ErbB in response to PI3K or AKT inhibition has also been well described [29,30]. Whether more direct co-targeting of MEK/ERK and PI3K/AKT may be a better strategy for some subsets of cancers has yet to be clearly established and it is an area of active investigation [44,45,46,47].

Our study is not without limitations. The models that are reported herein are based on PCa cell lines that do not express AR, which in general continues to be expressed in the majority of patients with progressive PCa, including in the treatment-refractory disease. Both forward and feedback interactions exist between the ErbB family, AR, AKT and ERK, which introduces another level of complexity with respect to intracellular signaling [48,49,50,51,52,53,54]. These signaling loops highlight the complex biology that is driving PCa, but also provide opportunities for potential therapeutic interrogation. Therefore, it remains relevant to continue to evaluate the role of combinatoric targeting of ErbB, AR and/or AKT in PCa, as has also been suggested by other investigators [48,49,50,51,55,56,57].

## 4. Materials and Methods

### 4.1. Cell Lines and Drugs

DU145 and PC3 human prostate cancer (PCa) cells were obtained from the American Type Culture Collection (ATCC) and confirmed by short tandem repeat (STR) analysis (Appendix A). The cells were cultured in Dulbecco’s Modified Eagle’s Medium (DMEM)/F12 medium (Invitrogen, Carlsbad, CA, USA) that was supplemented with 5% fetal bovine serum (Gemini, West Sacramento, CA, USA). Docetaxel- and paclitaxel-resistant DU145 and PC3 cells were derived from their respective wild-type DU145 and PC3 parental lines through a stepwise selection in increasing concentrations of docetaxel or paclitaxel over time, as described previously [33]. The final drug selection concentration in culture for the docetaxel-resistant DU145 cells, designated DU145/Doc, was 60 nM docetaxel, while for the paclitaxel-resistant PC3 cells, designated PC3/Pac, it was 20 nM paclitaxel, respectively. The selected cells were maintained in the drugs in culture up to the time of the relevant assays. Paclitaxel (Bedford Laboratories, Bedford, OH, USA) and docetaxel (Sanofi-Aventis, Bridgewater, NJ, USA) stock solutions were a gift from the University of Maryland Greenebaum Comprehensive Cancer Center (UMGCCC) Pharmacy. MK2206 was a gift from Merck (Kenilworth, NJ, USA), and lapatinib was purchased from Biovision Inc. (Milpitas, CA, USA). Stock solutions (50 mM) of both MK2206 and lapatinib were prepared in sterile DMSO and stored in −20 °C. The stock solutions were stable for more than one year. Working concentrations were freshly prepared before their use.

### 4.2. Cell Growth Inhibition MTT Assays

The 3-3-(4,5-dimethylthiazol-2-yl)-2,5-diphenyltetrazolium bromide (MTT) (Sigma-Aldrich, St. Louis, MO, USA) assay was used to assess the anti-proliferative effects of various drugs. Cells were plated in triplicates at a density of 2000–3000 cells per well in 96-well plates for 24 to 48 h prior to drug treatments. To determine the IC50 values, the cells were treated with a range of drug concentrations for 72 h, and the MTT was added, and the resulting formazan crystals were dissolved in isopropanol, and optical density was determined at 570 nm, as described in Xie et al. [58]. The IC50 values were calculated using GraphPad Prism 9.1.0 software (GraphPad Software, San Diego, CA, USA). To assess for relative anti-proliferative effects of the drugs at different time points, the cells that were seeded in triplicates in 96-well plates were treated with fixed doses of lapatinib (3 µM) or MK2206 (6 µM for PC3, PC3/Pac and 15 µM for DU145, DU145/Doc) or both for 1, 2 or 5 days, and then analyzed at each time point via the MTT assay. The standard error of mean (SEM) for each treatment was determined from the respective triplicate samples; *p*-values were determined using Student’s *t*-test.

### 4.3. Drug Combination Studies

These were performed as described by Chou and Talalay [59]. Specifically, two-drug combinations were evaluated via the MTT assay by combining the drugs at their fixed IC50 ratios and treating the cells over a range of serial dilutions for three days. The combination index (CI) values at 50, 75, and 90% effective doses for each drug in the combination were calculated using the CalcuSyn software version 2.11 for Windows (Biosoft: Chou, 1996–2012), with the drug combinations being considered as synergistic if the CI was <1, additive if the CI was between 1 and 1.2, and antagonistic if the CI was above 1.2 [36,59]

### 4.4. Western Blots

Western blots were performed as described previously [58]. Briefly, following the indicated treatments, the cells were washed twice in ice cold phosphate buffer saline (PBS), the attached cells were then scraped and lysed in RIPA buffer (1% NP-40, 0.5% Na2deoxycholate, 1% SDS, 5 mM EDTA in PBS) with 1 ± Protease and Phosphatase Inhibitor cocktail (Sigma-Aldrich, St. Louis, MO, USA) for 30 min on ice with occasional vortexing. The lysed cells were centrifuged at 14,000 rpm at 4 °C for 10 min. The protein concentrations in the supernatant were measured with Pierce BCA Protein Kit (Thermo Fisher Scientific, Waltham, MA, USA). Equal amounts of protein were run through SDS-PAGE 12% Bis-Tris gels (ThermoFisher Scientific, Waltham, MA, USA) and transferred onto Immobilon PVDF membranes (Millipore Sigma, Burlington, MA, USA) using the SureLock Novex blotting kit (ThermoFisher Scientific). The membranes were treated with 5% non-fat milk solution for 1 h at room temperature (RT) to block non-specific protein binding, washed and treated overnight with the primary antibodies of interest, followed by peroxidase-labeled secondary antibodies (anti-mouse or anti-rabbit; SeraCare, Milford, MA, USA) for 60 min. The primary antibodies used in the study (Appendix A) were obtained from the following sources—Cell Signaling (Beverly, MA, USA): pAKT, AKT, pERK, ERK, pEGFR, EGFR, pErbB2, pErbB3, ErbB3, PARP or cleaved PARP, BCL2, BAX, GAPDH; Proteintech (Rosemont, IL, USA): ErbB2; Sigma-Aldrich (St. Louis, MO, USA): PGP, BCRP; Santa Cruz Biotechnology Inc. (Santa Cruz, CA, USA): MRP1. The bands representing the different proteins were visualized using the ECL kit (KwikQuant, Kindle Biosciences, Greenwich, CT, USA), photographed using a digital camera and quantified using ImageJ software (Rasband, W.S., ImageJ, U.S. National Institutes of Health [NIH], Bethesda, MD, USA). For each representative Western blot that is shown, the densitometry data and SEM were obtained from at least two or three independent blots.

### 4.5. Cell Cycle Distribution Analysis

Cells (5 × 10^5^) were plated in 6-cm dishes in DMEM/F12 medium that was supplemented with 5% FBS for 24 h, then treated with DMSO (control), lapatinib (3 µM), MK2206 (6 µM for PC3 and PC3/Pac cells and 15 µM for DU145 and DU145/Doc cells) or both agents for 24 h. In other experiments, the PC3 cells were treated with paclitaxel (20 nM) and DU145 cells with docetaxel (60 nM) for 24 h before their harvest for a cell cycle analysis. Only the attached cells were used for a cell cycle analysis. Cells were trypsinized, washed twice in PBS, and fixed in 70% ice-cold ethanol. The cells were stained with staining buffer (50 µg/mL propidium iodide, 100 µg/mL RNaseA in PBS) at 37 °C for 30 min. Cell cycle profiles and distributions were determined by flow cytometry using the BD FACS Canto II Flow Cytometer (BD Biosciences, Franklin Lakes, NJ, USA) and FACS DIVA software V 8.0.1 (BD Biosciences). The SEM was determined from three independent experiments for each treatment.

### 4.6. Apoptosis Assay

The assays were performed using the BD Pharmigen Apoptosis Detection Kit (BD Biosciences). Cells were treated for 24 h or 48 h with drugs, then both the floating and attached cells were collected, rinsed in PBS and labeled with FITC-labeled PI according to the manufacturer’s instructions. The cells were analyzed via the BD FACS Canto II Flow Cytometer and the data processed using BD FACS DIVA software. The SEM was determined from at least 3 independent experiments.

### 4.7. Live/Dead Assay

Cells were plated in 6-cm dishes in DMEM/F12 medium that was supplemented with 5% FBS 24 h prior to drug treatment. The cells were then treated with DMSO (control), lapatinib (3 µM), MK2206 (6 µM for PC3-lineage cells, 15 µM for DU145-lineage cells) or both for 72 h. The attached cells were collected by trypsinization, re-suspended in complete medium at a concentration of 1 × 10^6^/mL. Acridine orange (AO) and PI reagents (Nexcelom Bioscience, Lawrence, MA, USA) were added to the cells per the manufacturer’s instructions and loaded into a counting chamber. Both live cells (AO positive, green fluorescence) and dead cells (PI positive, red fluorescence) within the counting chamber were assessed using a Nexcelom Cellometer Fluorescent Viability Cell Counter (Nexcelom Bioscience). The data were analyzed using FCS Express 6 software (De Novo Software, Glendale, CA, USA). SEM was calculated from two independent experiments.

### 4.8. Two Dimensional Clonogenic Assay

Cells/well (2 × 10^3^) were plated in triplicates in 12-well plates. After 48 h, cells were treated with DMSO (control), lapatinib (3 µM), MK2206 (6 µM for PC3-lineage cells, 15 µM for DU145-lineage cells) or both drugs and incubated for 10 days (media containing the drugs were changed on days 3 and 6). The media was aspirated thereafter, and the colonies washed with PBS and fixed with 200-proof ethanol for 30 min. After removing the ethanol, the colonies were stained with 0.5% crystal violet for 30 min at RT. After washing the wells with water, the violet-stained attached colonies were photographed. The surface area of the wells covered by the colonies was assessed using ImageJ 1.48v software (NIH, Bethesda, MD, USA). The results are presented in tabular format. The SEM was determined for each treatment from three different wells. A Student’s *t*-test was performed to determine the *p*-values.

### 4.9. In Vivo Studies

All of the animal studies were conducted under a protocol that was approved by the University of Maryland School of Medicine IACUC (American Association for Laboratory Animal Science). Initial pilot dose-finding studies using parental DU145- and PC3 cell-based tumor xenografts in male nude mice (Envigo, Frederick, MD, USA) that were 6 weeks of age were conducted using the published literature as a guide [31,60,61,62,63,64]. Based on these studies, the dose for lapatinib was determined to be 80 mg/Kg M-F for either 4 or 5 consecutive weeks, for MK2206 96 mg/Kg M, W, F for 4 or 5 consecutive weeks, and for the two drug combinations 80 mg/Kg lapatinib M-F + MK2206 96 mg/Kg M, W, F for 4 or consecutive weeks; for the combination studies, lapatinib was given first followed 4 h later by MK2206. All of the drugs were given using an oral gavage. Control mice received 0.2 mL sterile vehicle formulation per dose daily M-F for 4 or 5 consecutive weeks using an oral gavage. For the efficacy studies, PC3 or DU145 cells (70–80% confluent) were collected by trypsinization, rinsed and re-suspended in PBS, admixed with 33% phenol red free matrigel (Corning, NY, USA), followed by subcutaneous injection of the admixture (100 µL/injection) into the flanks of the mice (one injection per mouse)—for PC3 cells, 5 × 10^6^ cells/mouse and for DU145 cells, 5 × 10^6^ cells/mouse were injected per flank. Tumors formed within 4 weeks with an average size of 120 mm^3^. The mice were sorted into four treatment groups, with 7–8 mice per group: vehicle, MK2206 alone, lapatinib alone, or MK2206 + lapatinib. Mouse body weights were measured M-F of each week which served as a surrogate for tolerability and toxicity, with more than 10% loss in total body weight being considered as an indicator of dose-limiting toxicity. Tumor length and width were measured twice a week using Traceable Digital Calipers (VWR, Radnor, PA, USA). Tumor volumes were calculated using the following formula: 4/3πa2b where a = 0.5 width; b = 0.5 length (a < b). At the end of approximately 30 days, the mice were euthanized, and the harvested tumors were weighed and frozen for subsequent studies (the Western blots).

### 4.10. Statistical Analysis

Comparisons between the different treatment groups were done by a Student’s *t*-test using an Excel program (MS Excel). GraphPad Prism 9.1.0 was also used to prepare graphs, calculate standard error of means (SEM) and to perform a Mann-Whitney U test.

## 5. Conclusions

The present study suggests a possible strategy for targeting aggressive PCa, including potentially aggressive variant, androgen-independent and chemotherapy-refractory phenotypes. Next generation ErbB receptor and AKT targeting agents with improved activities and hopefully fewer toxicities, and the better identification of the subsets of the PCa patients that are most likely to benefit from well-defined signal pathway modulation are necessary to successfully extend such approaches to patients with advanced and progressive PCa [65,66,67]. Given that ErbB signaling can be tumor-permissive by negatively impacting anti-tumor immunity, it will also be of interest in future studies to determine whether ErbB/AKT co-targeting can be integrated with immune checkpoint inhibition to improve the treatment outcomes in PCa, which generally otherwise is minimally responsive to an immune checkpoint blockade [68].

## Figures and Tables

**Figure 1 cancers-14-04626-f001:**
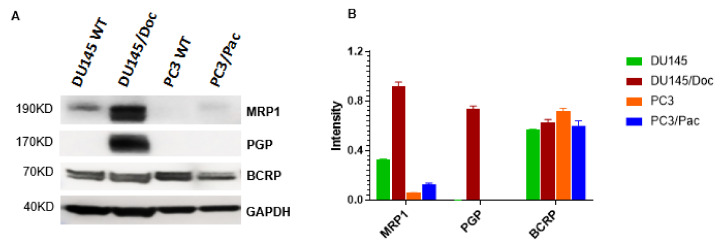
ATP-Binding Cassette Transporter Protein Expression (MRP1, PGP, BCRP) in wild-type and taxane-resistant prostate cancer cells. (**A**) Western blot. Thirty-five ug total protein samples were loaded per lane. The uncropped blots are shown in Appendix A. (**B**) Densitometry. Summary of results from three independent Western blots.

**Figure 2 cancers-14-04626-f002:**
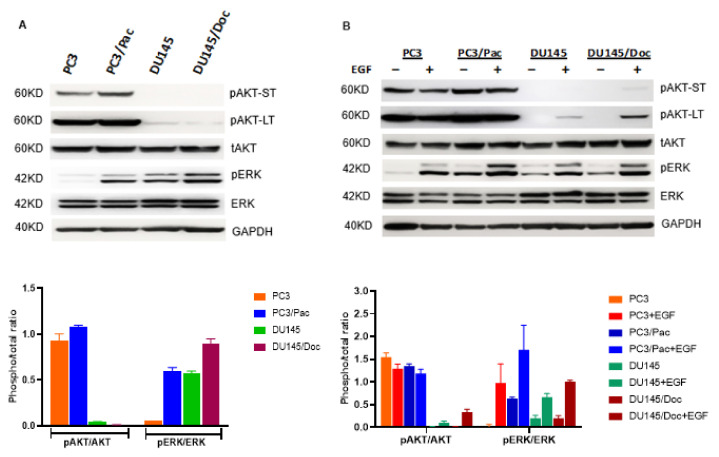
ErbB Axis in wild-type and taxane-resistant prostate cancer cells. These are Western blots. Each lane has 45 ug total protein. (**A**) Basal pAKT, pERK expression in regular cell culture growth conditions (5% FBS). Both short-term (ST, 2 min) and longer term (LT, 6 min) film exposures are shown for pAKT. The pAKT/total AKT and pERK/total ERK densitometry ratios are from two independent blots. (**B**) pAKT, pERK expression in response to EGF treatment. Cells were serum starved (0.5% FBS) overnight, then treated with EGF (50 ng/mL, 10 min) before the cell harvest. Densitometry ratios are from two independent blots. (**C**) Time course of ErbB-mediated downstream signaling in PC3 and PC3/Pac cells in response to EGF treatment (50 ng/mL). Cells were serum starved (0.5% FBS) overnight, then treated for the indicated times with EGF before the cell harvest. (**D**) Time course of ErbB-mediated downstream signaling in DU145 and DU145/Doc cells (serum starved overnight) in response to the EGF treatment (50 ng/mL). The uncropped blots are shown in Appendix A.

**Figure 3 cancers-14-04626-f003:**
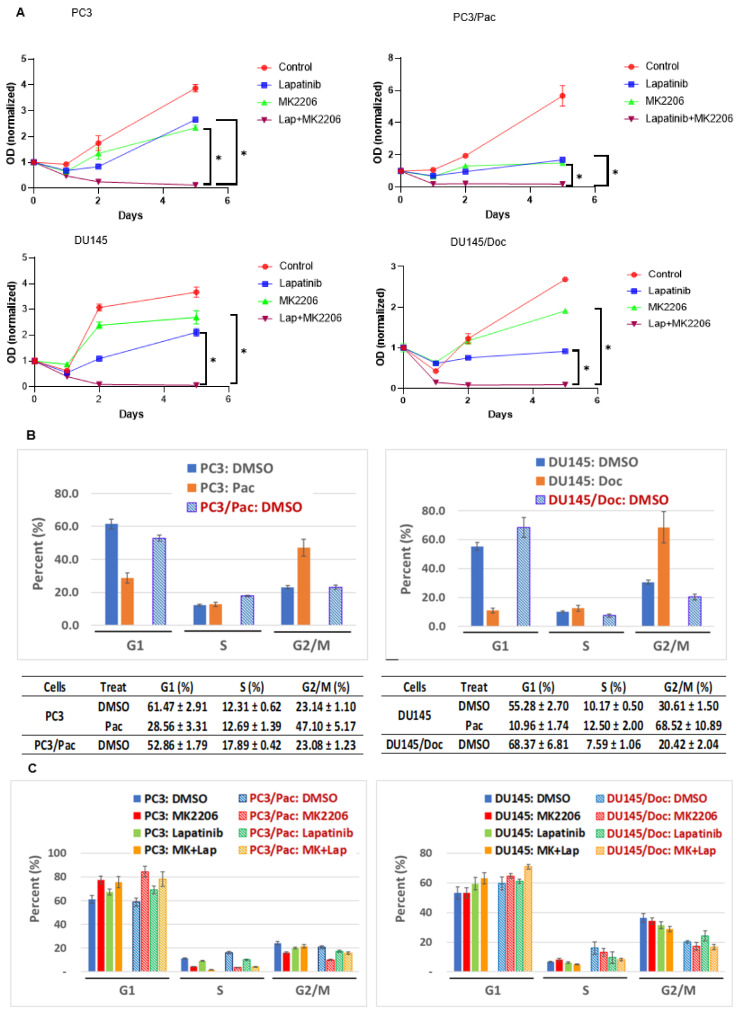
Effects of ErbB and AKT targeting on cell proliferation and cell death. The doses of the inhibitors that were used for all experiments, either singly or in combination, were lapatinib 3 µM for all four cell lines, 6 µM MK2206 for PC3, PC3/Pac and 15 µM MK2206 for DU145, DU145/Doc. A DMSO treatment served as a control. (**A**) MTT assays. Anti-proliferative effects at different time points of treatment. Cells that were seeded in 96-well plates in triplicates were treated with the respective agents for days 1, 2 and 5, and analyzed by an MTT assay at each time point. Experiments were repeated three independent times. * *p*-value < 0.05. (**B**) Cell cycle analysis. Response of the cells to the taxanes. Wild-type PC3 and DU145 cells were seeded in 6-well plates ON, then treated with DMSO (control), paclitaxel 20 nM or docetaxel 60 nM, respectively, for 24 h before analysis. PC3/Pac and DU145/Doc cells were maintained in paclitaxel or docetaxel in culture till time of cell cycle analysis. Results are summarized from three independent experiments. (**C**) Cell cycle analysis. Response of cells to lapatinib and MK2206. Wild-type and taxane-resistant cells were treated with lapatinib, MK2206, or both, for 24 h before analysis. Results are summarized from three independent experiments. (**D**) Assessment of apoptosis by flow cytometry. Wild-type and taxane-resistant cells were treated with DMSO, MK2206 or both for 48 h, then stained with annexin V and PI prior to analysis by flow cytometry. Representative data from flow cytometry are shown, and total (early + late) apoptosis for each treatment are summarized from three independent experiments. (**E**) Live-dead assay. Cells plated in 6-cm dishes were treated with the respective agents as shown for 72 h, stained with acridine orange (AO) and PI, and assessed for whether they were live (AO positive, green fluorescence) or dead (PI positive, red fluorescence) using a Nexcelom Cellometer Fluorescent Viability Cell Counter. Data analyzed using the FCS Express 6 software are summarized from two independent experiments. (**F**,**G**) Western blots. PARP, BCL2, BAX expression. Cells were treated with DMSO, lapatinib, MK2206 or both agents for 24 h, then whereas indicated with EGF (50 ng/mL, 10 min) before cell harvest. Forty-five ug protein samples were loaded per well. BAX/BCL2 densitometry ratios from two independent blots are also shown. The uncropped blots are shown in Appendix A.

**Figure 4 cancers-14-04626-f004:**
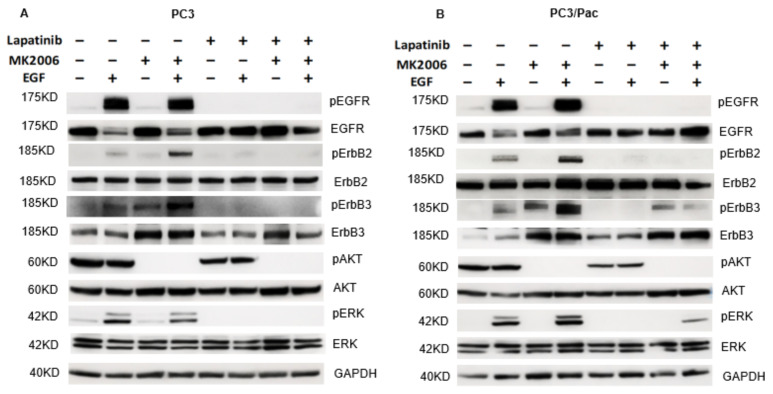
ErbB and downstream signaling in PC3 and PC3/Pac cells. Western blots. Cells were treated with DMSO, lapatinib (3 µM), MK2206 (6 µM) or both agents for 24 h, then they were indicated with EGF (50 ng/mL, 10 min) before the cell harvest was conducted. Forty-five ug protein samples were loaded per well. (**A**) PC3, and (**B**) PC3/Pac. The uncropped blots are shown in Appendix A.

**Figure 5 cancers-14-04626-f005:**
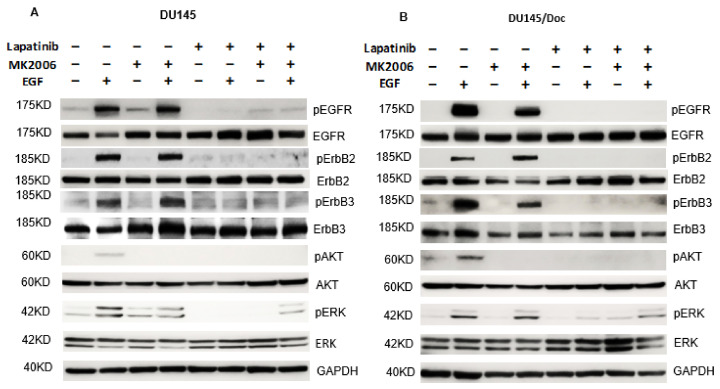
ErbB and downstream signaling in DU145 and DU145/Doc cells. Western blots. Cells were treated with DMSO, lapatinib (3 µM), MK2206 (15 µM) or both agents for 24 h, then, where indicated, with EGF (50 ng/mL, 10 min) before the cell harvest was conducted. Forty-five ug protein samples were loaded per well. (**A**) DU145, and (**B**) DU145/Doc. The uncropped blots are shown in Appendix A.

**Figure 6 cancers-14-04626-f006:**
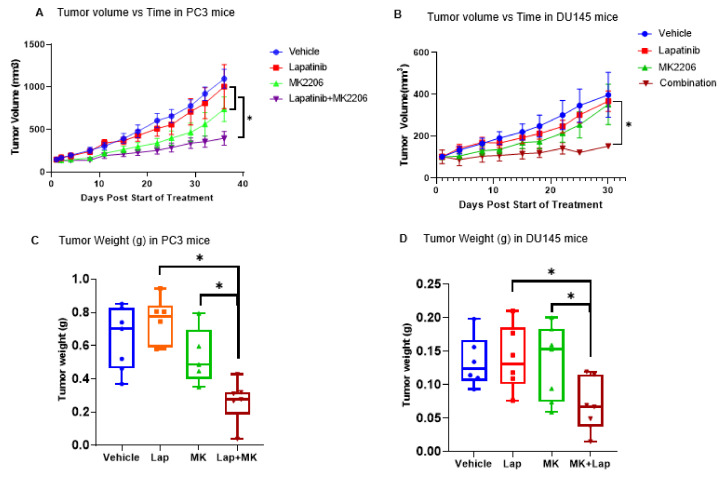
Targeting ErbB and AKT in vivo. Evaluation of PC3- and DU145-bearing tumor xenografts. (**A**,**B**) Tumor volumes of PC3 (**A**) and DU145 (**B**) tumors over time in response to the indicated treatments. Tumor volumes were assessed twice a week over the course of approximately 30 days. Comparison of the differences between the tumor groups was determined by a Mann-Whitney U test. * *p*-value < 0.5. (**C**,**D**) Weights (in grams) of PC3 (**C**) and DU145 (**D**) tumors that were harvested at the end of the experiment. * *p*-value < 0.5. (**E**,**F**) Western blots. Sixty ug protein samples from five independent tumor specimens for each treatment group were evaluated by Western blots for pAKT/AKT and pERK/ERK expression. Group I, vehicle-treated; Group II, lapatinib-treated; Group III, MK2206-treated; Group IV, lapatinib + MK2206-treated. The uncropped blots are shown in Appendix A.

**Table 1 cancers-14-04626-t001:** IC50 values for wild-type and taxane-resistant cells.

	PC3	PC3/PAC	DU145	DU145/DOC
MK2206 (IC50 µM)	5.5 ± 0.5	6.4 ± 0.4	13.5 ± 1.5	16.1 ± 0.9
Lapatinib (IC50 µM)	2.75 ± 0.25	3.05 ± 0.31	2.65 ± 1.35	1.8 ± 0.2
Paclitaxel (IC50, µM)	0.018 ± 0.003	0.2 ± 0.015	0.0025 ± 0.00015	0.25 ± 0.5
(fold-resistance)		(11-fold)		(100-fold)
Docetaxel (IC50, µM)	0.0008 ± 0.0001	0.0024 ± 0.0002	0.0005 ± 0.00015	0.25 ± 0.009
(fold-resistance)		(3-fold)		(500-fold)

PC3 is PTEN negative, p53 null; DU145 cells have wild-type PTEN, mutant p53.

**Table 2 cancers-14-04626-t002:** Combination Index (CI) values of MK2206 and Lapatinib for wild-type and taxane-resistant cells.

Cell Lines and Fraction Affected	Combination Index
**PC3**	
0.5	0.75279
0.75	0.69323
0.9	0.63849
**PC3/Pac**	
0.5	0.8483
0.75	0.67993
0.9	0.54582
**DU145**	
0.5	0.51149
0.75	0.62246
0.9	0.82236
**DU145/Doc**	
0.5	0.66591
0.75	0.74173
0.9	0.89268

## Data Availability

The data presented in this study are available in this article and Appendix A.

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
