# Peer review of "Co-Targeting ErbB Receptors and the PI3K/AKT Axis in Androgen-Independent Taxane-Sensitive and Taxane-Resistant Human Prostate Cancer Cells"

_cancers, 2022, doi:10.3390/cancers14194626_

Round 1

Reviewer 1 Report

In this well-written paper, the authors focused on PC3 and DU145 cells that are AR negative so androgen-independent pathway could be studied having several similarities with NED clones.

The main rationale is that the combination treatment based on novel drugs targeted at ErbB receptors and downstream effectors such as PI3K or AKT has the potential to result in more intense anti-tumor effects than monotherapy. Thus, the authors used both wild-type and taxane-resistant PC3 and DU145 cells as models of aggressive androgen-independent and chemotherapy-resistant PCa to assess the role of ErbB receptors- and AKT-orientated treatment under basal conditions and in response to pathway activation. The experiments were well-designed and the results are clearly presented.

Some minor comments are as follows:

1.     In many figures resolution needs to be upgraded, especially as for the incorporated tables

2.     Line 73: is there are role for immunotherapy in AVPC?

3.     Lines: 106-118 there is no need to repeat the abstract/results in introduction. All in all, the introduction should be shortened. 

4.     Lines 125-132: these lines can be used in the discussion rather than as the repetitive introduction for the results. 

Author Response

We are grateful for the very constructive comments. Based on our reviewers comments we have made specific changes as outlined below. Two copies of the revised manuscript are enclosed, one with highlighted changes/revisions, and one clean copy for your kind consideration. Please note the line numbers of the original manuscript have changed in the updated manuscript due to the requested revisions.

  1. We have gone back and carefully reviewed all the figures and Tables to improve the presentation. This comment was made by both Reviewer 1 and Reviewer 2, particularly for Figure 3 - we have updated the figures and separated out the incorporated Tables to improve the presentation and remove any figure overlaps.
  2. To date, there is no role for immunotherapy for patients progressing on hormone therapy/chemotherapy which we have now stated in the Introduction, second paragraph.
  3. As suggested, we have removed lines 102-119 in the revised version – this includes last few lines of the Introduction as well as the first paragraph of the Results Section 2.1 to minimize redundancy with the Abstract/Intro/Results section.
  4. We have also removed the third paragraph from the Results Section 2.1 (lines 135-145). This is now moved to the Discussion Section (new paragraph 4 of Discussion).

Due to the above changes, some of the reference numbering has also changed and these are also highlighted for clarity. We again want to thank our reviewers and believe the revisions have further enhanced the flow of the paper.

Reviewer 2 Report

In this paper, Adediran et al. explore co-targeting of ErbB Receptors and the PI3K/AKT axis in androgen-independent taxane-sensitive and taxane-resistant human prostate cancer cells. The authors show that co-targeting AKT with ErbB may be a useful strategy to explore further for potential therapeutic effect in advanced prostate cancer.

General comment:

The experiments are well designed and the paper is well written.

Specific points:

Is there a possibility to confirm the results obtained with inhibitors with siRNAs or shRNAs?

Figures 3b and 3d are little bit blurred.

Figure 3e, 3f, 3g: WBs and IF images are overlapping.

Section 4.4.: It would be good to write the catalog numbers of antibodies and the dilution that has been used.

Author Response

We are grateful for the very constructive comments. Based on our reviewers comments we have made specific changes as outlined below. Two copies of the revised manuscript are enclosed, one with highlighted changes/revisions, and one clean copy for your kind consideration. Please note the line numbers of the original manuscript have changed in the updated manuscript due to the requested revisions.

  1. We thank the Reviewer about the siRNA/shRNA suggestion. We did consider this originally, and although off target effects are always a consideration with most pharmacologic agents, given that our focus was to have translational relevance using strategies (with small molecule kinase inhibitors for example) that have the potential to be extended to clinical practice, and given that we carefully evaluated the effects of these inhibitors on the respective targets via multiple western blots that demonstrated the expected pharmacodynamic target effects, we believe we have been able to draw the conclusions presented in the current paper.  
  2. We have revised Figure 3 for better clarity and redone the spacing to remove any overlaps between the figures.
  3. We have added a new Supplementary Table 2 with the antibodies, their catalog numbers and dilutions used in the experiments.

Due to the above changes, some of the reference numbering has also changed and these are also highlighted for clarity. We again want to thank our reviewers and believe the revisions have further enhanced the flow of the paper.